# Phenolic, Nutritional and Sensory Characteristics of Bakery Foods Formulated with Grape Pomace

**DOI:** 10.3390/plants13050590

**Published:** 2024-02-22

**Authors:** Andrea Antoniolli, Lucía Becerra, Patricia Piccoli, Ariel Fontana

**Affiliations:** 1Cátedra de Química Orgánica y Biológica, Departamento de Biomatemática y Fisicoquímica, Facultad de Ciencias Agrarias, Universidad Nacional de Cuyo (UNCuyo), Chacras de Coria M5528AHB, Argentina; aantoniolli@fca.uncu.edu.ar (A.A.); mabec.lucia@gmail.com (L.B.); ppiccoli@fca.uncu.edu.ar (P.P.); 2Grupo de Bioquímica Vegetal, Instituto de Biología Agrícola de Mendoza (IBAM), Consejo Nacional de Investigaciones Científicas y Técnicas (CONICET), Universidad Nacional de Cuyo (UNCuyo), Almirante Brown 500, Chacras de Coria M5528AHB, Argentina

**Keywords:** anthocyanins, phenolic compounds, wine industry by-products, bakery functional foods, antioxidant dietary fiber

## Abstract

The potentiality of cv. Malbec grape pomace (GP) as a functional ingredient in the formulation of bakery foods (muffins, biscuits and cereal bars) was studied. The effect of GP addition on the phenolic compounds (PCs) composition, nutritional and sensory properties were evaluated. The addition of GP increased the content of dietary fiber, proteins, ash, total phenolic content (TPC), antiradical capacity (AC), anthocyanins and non-anthocyanin PCs while decreasing the carbohydrates content. The main PCs given by the GP to supplemented foods were quercetin-3-*O*-glucoside, rutin, caffeic acid, syringic acid and (+)-catechin. For anthocyanins, the acylated derivatives were more stable to heat treatment (baking) in food processing which was evidenced by a higher proportion of these PCs compounds when compared to the same derivatives quantified in GP. In general, when the TPC or individual concentrations of PCs were analyzed in a nutritional or functional context, one portion of the supplemented foods showed levels high enough to satisfy the recommended dose per day of these bioactive compounds. Additionally, the foods were well received by consumers during the sensory evaluation and supplemented biscuits received the highest acceptability. This study demonstrated that GP could be a viable functional ingredient in bakery foods to incorporate components like PCs and dietary fiber into traditional consumers’ diets.

## 1. Introduction

The wine industry annually generates large amounts of solid wastes with high seasonal disposal dependence [1]. According to previous reports, the winemaking step is one of the most relevant in terms of by-product generation and also from the point of view of its potential economic importance. In this step, grape pomace (GP) is the most abundant by-product that accounts for, on average 30%, of processed grapes before winemaking. Traditional exploitation of this material was mainly based on the extraction of ethanol, tartaric acid and seed oil, but these basic applications still retain in GP some under-exploited components. In this sense, the better use of other constituents of this by-product for novel products has been explored by performing different technological applications [2].

GP is a rich source of bioactive constituents, including PCs, proteins, lipids, soluble and insoluble fiber and minerals. Therefore, new possibilities for a potential second life by its sustainable reuse have emerged. Bioactive components of GP have been evaluated, focusing on their composition and biological properties [3]. Furthermore, PCs and their association with antioxidant and antimicrobial activity as well as their interaction with other food ingredients, such as the functional fiber and grape seed fatty acids, have been reported [4]. Likewise, some previous research from our group studied the composition of Malbec GP, for which high concentrations of PCs, dietary fiber and antioxidant activities were found and later associated with health benefits in animal experimental models [5,6]. Taking into account these previous reports related to the improvement of pathological situations, food products containing GP or its derived bioactive constituents can be potentially considered as functional foods. Interest in by-products rich in PCs has been initially focused on the antiradical capacity (AC) resulting from their chemical structure with several hydroxyl groups [7]. Moreover, being 43–75% of GP’s content, dietary fiber is the most important constituent of GP and has been associated with a reduction in the risk of cardiovascular diseases, diabetes prevention, cancer protection, cholesterol levels decrease and constipation and obesity prevention [8,9]. The high fiber content of GP, consisting mainly of glycans, cellulose and pectins, emphasizes its nutritional value by providing a source of bioactive components in the formulation of functional foods [10,11]. Malbec GP has been reported to have a high fiber content (53%) and this composition has been associated with lower blood pressure and triglyceride levels, evidenced by an improvement in the insulin resistance of rats fed with GP [6]. GP dietary fiber is considered a matrix rich in PCs, being a dietary supplement that combines, on one hand, the inherent benefits of dietary fiber and, on the other hand, its richness in bound PCs, also defined as non-extractable polyphenols (NEPPs) with antioxidant and microbiota modulation properties [12]. The antioxidant effect of dietary fiber is based on PCs bound to polysaccharide complexes, which are released in the intestine and act as antioxidants or microflora modulators [13]. Thus, GP has potential as a raw material for the production of dietary fiber concentrates and the development of commercial fiber-rich products with added nutritional value through the content of PCs [14]. Other important components of GP include proteins, polysaccharides and oligosaccharides [15]. The protein content of GP can vary between 6% and 15%. The amino acids profile is comparable to that of cereals, with high levels of glutamic and aspartic acids and low levels of tryptophan and sulfur-containing amino acids [16]. In terms of mineral content, it is variable and the most abundant minerals are potassium tartrates, with contents ranging from 4 to 14% of dry matter [17]. The main fatty acids are linoleic (C18:2), oleic (C18:1) and palmitic (C16:0) acids with contents of around 70, 15 and 7%, respectively [15]. ß-sitosterol represents the main sterol in grapeseed oil and α-tocopherol was reported as the main tocopherol [18].

Previous studies have evaluated the addition of GP and grape peels from different white and red varieties in different baked goods including bread, cookies, muffins, brownies and cakes [19,20,21,22,23,24,25]. Some of these studies evaluated the effects of the GP addition on the nutritional, sensory and chemical composition of finished products, highlighting the potential of GP as a way to incorporate bioactive components like PCs and fiber.

In this context, the aim of the present study was to investigate the effect of the addition of dried cv. Malbec GP in the formulation of muffins, cereal bars and biscuits to incorporate bioactive components of natural origin like PCs and dietary fiber. The nutritional and antiradical capacity (PCs composition and ACs) were evaluated to understand the potentiality of GP as a sustainable source of active dietary fiber and bioactive phytochemicals supplementation. Additionally, a sensory analysis of control and fortified foods was performed to estimate the acceptability of the products by consumers.

## 2. Results and Discussion

### 2.1. Total Phenolic Content and Antiradical Capacity

TPC and AC provide valuable information on the relative composition of samples and their potential as functional foods. Table 1 summarizes the results obtained for the determination of these parameters. The data obtained showed an increase in TPC in foods supplemented with GP compared to the control, and it was proportional to the different percentages of GP incorporation in each food. The greatest difference between the witnessed food and that elaborated with GP was observed for muffins, which increased the TPC more than 8 times. Then, the biscuits and cereal bars showed increases in TPC of 6 and 3 times, respectively.

The results for TPC are in agreement with those reported by Mildner-Szkudlarz et al. [26]. The authors carried out a study on cookies by adding white GP. Mildner-Szkudlarz et al. observed that by adding 10% of GP to the final product, 2.11 mg GAE g^−1^ DW was obtained compared to the product without GP (0.85 mg GAE g^−1^). With the addition of 20%, they obtained 3.34 mg GAE g^−1^; and with 30%, 4.45 mg GAE g^−1^, representing increases of 148%, 293% and 424%, respectively. Hayta et al. [22] prepared bread with up to 10% GP powder and obtained 0.89 compared to 0.35 mg GAE g^−1^ DW in the control (2.5-fold increase). Šporin et al. [19] added 15% Merlot and Zelen GP, obtaining 5.92 and 3.65 mg GAE g^−1^ DW, respectively, compared to less than 0.69 in the control bread. An increase from 0.19 in the control to 0.54 mg GAE g^−1^ DW in cakes with 10% GP was reported by Nakov et al. [23]. Walker et al. [25] reported that the highest TPC values for bread and muffins were obtained in the 15% Pinot Noir GP-fortified samples, with values (expressed as mg GAE per serving) of 22 and 512%, respectively. The control brownies had higher TPC values than the fortified samples, probably due to the high content of chocolate. If we compare the previous studies with the results obtained in this work, it can be observed that the TPC values are higher than those described previously. A possible explanation could be related to the higher content of PCs in the Malbec GP (information presented in the next section).

The above-mentioned increases were associated with the AC results. Among the studied foods, the highest increase in AC was observed in muffins (6 times), followed by biscuits (4 times) and cereal bars (3 times) for the same added proportion of GP (Table 1). Our results agree with those reported by Mildner-Szkudlarz et al. [26] in similar kinds of foods. Mildner-Szkudlarz et al. obtained 3.39 μmol TE g^−1^ DW in biscuits with 10% addition of GP while, with 20% addition, they obtained 5.12 μmol TE g^−1^ DW and, with 30% addition, 7.55 μmol TE g^−1^ DW. Likewise, in biscuits without addition of GP, they found 1.27 μmol TE g^−1^ DW; thus, the increases were 3, 4 and 6 times that of the control, respectively.

### 2.2. Characterization of Phenolic Compounds Composition

#### 2.2.1. Anthocyanins

The quantification results for the identified anthocyanins in the GP-supplemented foods are presented in Table 2. The non-supplemented foods did not report the presence of any of the studied anthocyanin derivatives. A total of 10 glycosylated and acylated (acetyl and *p*-coumaroyl derivatives) anthocyanins were quantified in fortified samples. Malvidin 3-*O-p*-coumaroylglucoside was the predominant compound (68, 65 and 72% for muffins, cereal bars and biscuits, respectively), mostly followed by malvidin 3-*O*-glucoside in all the supplemented foods. This behavior was different to that observed in GP extracts of Malbec, where the malvidin 3-*O*-glucoside was the most abundant compound [5]. A potential explanation is the higher stability under high temperatures of acylated anthocyanin derivatives compared to non-acylated forms [27]. This indicates that the coumaryl-derived compounds prevailed after the heat treatment (baking) in food processing, while the non-acylated glycosides were more unstable. This result is also highlighted when the proportion of total acylated derivatives in foods compared to GP was calculated. In the GP-supplemented foods, the acylated anthocyanins content represented 76, 73 and 82% for muffins, cereal bars and biscuits compared to the 40% of these derivatives quantified in the same GP by Antoniolli et al. [5].

On the other hand, previous data suggested that the bioavailability of anthocyanins depends on the glycosidic substituent and, therefore, non-acylated anthocyanins are absorbed more efficiently than acylated ones [28]. In terms of anthocyanins distribution and changes presented and discussed in the present work, there are no previous publications studying in a detailed way the individual anthocyanins distribution in bakery foods supplemented with GP. Some of the previous publications supplementing GP in baked goods determined the total anthocyanin content [20,23], but the profiling and quantification of individual compounds are important in understanding the changes in the profiles. This information would be useful to know the remaining concentration of potential bioavailable compounds. In this sense, more studies evaluating the bioaccessibility and bioavailability of the remaining compounds after GP-supplemented food elaboration should be performed to understand the potential health beneficial effects of GP addition. 

Regarding the importance of anthocyanins as functional compounds, AC has been recognized. In fact, the European Food Safety Authority (EFSA) has made recommendations for consumers to obtain the expected health benefits of anthocyanins, suggesting a consumption of at least 20 to 40 mg per day [29]. For the supplemented foods under study, a muffin portion (55 g—as defined in the Code of Federal Regulations, 2022) [30] equates to 48.29 mg of anthocyanins per serving, a cereal bar (40 g) equates to 46.6 mg per serving and a biscuit (55 g) equates to 30.16 mg per serving. This indicates that by consuming a portion of a muffin or a cereal bar or two of the biscuits under study, the recommended daily dose for anthocyanins to exert a beneficial effect on health is covered. Despite the aforementioned assertions, the EFSA also mentions that anthocyanins consumption should not exceed the acceptable daily intake of 150 mg per day [29].

#### 2.2.2. Non-Anthocyanins

Table 3 presents the data of non-anthocyanins PCs quantified in the control and GP-supplemented foods. A total of 12 non-anthocyanin PCs from different chemical families, including non-flavonoids (hydroxybenzoic and hydroxycinnamic acids and stilbenes), flavonoids (flavanols and flavonols) and other compounds were identified and quantified in the extracts of GP-supplemented foods. Additionally, smaller numbers and concentrations of these compounds were found in control samples. The quantified PCs included OH-tyrosol, (+)-catechin, caffeic acid, syringic acid, *p*-coumaric acid, ferulic acid, chlorogenic acid, *trans*-resveratrol, quercetin-3-*O*-glucoside, rutin, myricetin and quercetin.

By individually analyzing the GP-supplemented foods, muffins showed quantifiable levels of 13 non-anthocyanin PCs, quercetin-3-*O*-glucoside being the most abundant PC, with concentrations between 132.1 and 322.8 μg g^−1^ (DW), followed by rutin, caffeic acid, syringic acid and (+)-catechin. In control muffins, only four PCs were found, with the following PCs being absent in this sample: syringic, caffeic and chlorogenic acids, *trans*-resveratrol, (+)-catechin, myricetin and OH-tyrosol. In the case of supplemented cereal bars, chlorogenic acid was the major compound, followed by (+)-catechin and rutin in decreasing concentrations. In the case of GP-supplemented biscuits, higher concentrations of quercetin-3-*O*-glucoside and caffeic acid were observed compared to their non-supplemented counterparts.

In general, in non-supplemented foods, OH-tyrosol, (+)-catechin, *trans*-resveratrol, *p*-coumaric, ferulic and chlorogenic acids were not found. In the study carried out by Antoniolli et al. [5], it was observed in Malbec GP that the flavan-3-ols were the most abundant compounds, with (+)-catechin, (−)-epicatechin gallate and (−)-epicatechin being the most abundant compounds. In the supplemented foods analyzed here, these compounds varied from one matrix to other; even some of those found in Malbec GP were not found in the food, as was the case for (−)-epicatechin gallate and (−)-epicatechin. The diminution or loss of flavanols like (+)-catechin and (−)-epicatechin could be related to their low stability under high temperatures like those of baking during food preparation. The loss of these compounds under high-temperature treatments was previously reported [31]. For stilbene-type compounds, *trans*-resveratrol was found in GP with a concentration of 2 μg g^−1^. This level agrees with those found in GP-supplemented foods, even being found in a higher proportion in biscuits and not detected in non-supplemented foods. By comparing the results obtained by Antoniolli et al. [5] and those of the present work, both indicated that qualitative and quantitative contents of the individual PCs can vary when GP is added to processed foods. Since it comes as an ingredient in supplemented foods, variables associated with the different accompanying matrices and processing technologies (baking temperature, cooking time and interaction with food components) could affect the presence of PCs in the final products. Another novel result found in this work is related to the content of *trans*-resveratrol in GP-supplemented foods.

According to the EFSA, an amount between 1 to 10 mg per day of *trans*-resveratrol should be consumed to produce a beneficial effect on health [29]. For the foods under study, the *trans*-resveratrol content in GP- supplemented samples was: 0.15 mg per portion (55 g) of muffins, 0.08 mg per portion (40 g) of cereal bars and 0.25 mg per portion (55 g) of biscuits. In this way, a portion of each of these foods, to a greater extent for biscuits, would contribute to reaching the recommended daily dose for this compound. In the same vein, the EFSA also establishes the recommended consumption of PCs from extracts (it does not establish criteria in specific foods), suggesting a minimum daily consumption of 300 mg of PCs [29]. In this study, the GP-supplemented foods contained an average of 142, 199 and 282 mg of GAE per portion of fresh product. Thus, they are mostly able to satisfy the recommended levels. 

### 2.3. Nutritional Properties 

The results of the chemical composition shown in Table 4 indicated that the most significant contribution (from a nutritional point of view) of the supplementation of foods with GP was the fiber content. In this respect, the biscuit product stands out with an increase of slightly more than 6 times compared to the control (from 0.9 g to 5.83 g/100 g FW), while cereal bars and muffins increased by 7 and 6 times, respectively.

Troilo et al. [32] added 15% GP powder with different particle sizes to muffins. They obtained between 3 and 3.9 g of total dietary fiber per 100 g of product. Nakov et al. [23] found that the addition of increasing amounts of GP powder (4 to 10%) in cakes gradually increased fiber content and other parameters, with total fiber increasing from 6.7 g (control) to 12.5 g of total fiber per 100 g DW (10% addition). In view of the observed results, the best option seems to be to incorporate GP and add a higher content of fiber.

According to FAO/WHO and Food and Drugs Administration [33,34] guidelines, the recommended daily intake of fiber is between 25 and 28 g/2000 kcal per day. The data presented here showed that the consumption of these foods supplemented with Malbec GP would increase the daily intake of fiber, helping to achieve the recommended daily dose and also improving the nutritional quality of baked goods. Furthermore, according to the EC Regulation 1924/2006, in order to claim that a food is a “source of fiber”, the product must contain at least 3 g of fiber per 100 g of food [35]. Both the GP-fortified cereal bars and the GP-fortified biscuits, with 3.31 and 5.83 g of fiber per 100 g FW, respectively, satisfy this requirement. The interest in including fiber in the diet, despite its indigestibility, is based on its beneficial effects on health. In this sense, it is important to provide it in an easy and practical way. Previous studies [36,37] have investigated the influence of fiber content on the glycemic index of foods, showing lower values of this parameter for foods with higher fiber content. So, the presence of GP, which leads to an increase in fiber content, could contribute to a reduction in the glycemic index of foods, although specific studies should be carried out to verify its influence. In addition, the proposed baked foods are an easy way to increase the amount of fiber in modern diets.

No differences were found in the protein content of the different foods (GP-supplemented foods compared with their control counterparts). Several studies have reported the protein content of GP. Paulos et al. [38] reported that white GP has a protein content of 10% (DW). Gazzola et al. [16] indicated that the protein content of GP can vary between 6% and 15% (DW) depending on the variety and growing conditions of vine plants. It should be noted that these values are similar to the protein content of the wheat flour used (10 to 12%), which could justify that the substitution of one ingredient for the other does not modify the final content of the fortified foods. Cereal bars were the only case where it was observed that the food with GP had a higher protein content. The reason could be that the GP was an additional ingredient in this food, causing a proportional increase in protein due to the lower protein content of the other ingredients used in this food, such as corn and rice flakes.

Regarding the protein quality of GP, there are few reports on this subject. However, in a study reported by Prandi et al. [39] related to the amino acid profile of GP, leucine (5.62 mg g^−1^) and phenylalanine (4.09 mg g^−1^) were quantified. In fact, the incorporation of GP could be an interesting alternative to improve the protein quality of supplemented foods. On the other hand, Gazzola et al. [16] showed that the amino acid profile of GP was comparable to cereals, with high levels of glutamic acid and aspartic acid and low levels of tryptophan and sulfur-containing amino acids. 

In terms of lipid content, there is an increased concentration in cereal bars and biscuits formulated with GP, while the control muffins have a higher lipid content (not statistically significant) than those formulated with GP. Although a lipid profile was not determined in the present study, data previously reported by Paulós et al. [38] suggest that the addition of GP to food systems would improve their lipid profile. According to the aforementioned work, GP is rich in linoleic acid (C18:2), corresponding to 68.78% of total fatty acids (FA), a higher value than sunflower, soybean or corn bagasse, which have linoleic acid contents of 65, 50 and 48% of total FA, respectively. Oleic acid (C18:1) is the second richest acid in GP (14% of the total AG). These two acids, linoleic and oleic, represent 83% of the total fatty acids in GP and about 90% in seeds. Similarly, Bordiga et al. [15] state that grapeseed oil is characterized by a high content of oleic and linoleic (unsaturated) acids. Furthermore, the lipid fraction is characterized by a fatty acid profile that shows a high amount of polyunsaturated/monounsaturated fatty acids and a low amount of saturated fatty acids. Linoleic (C18:2), oleic (C18:1) and palmitic (C16:0) acids are the main fatty acids in grapeseed oil, with percentages of approximately 70%, 15% and 7%, respectively [15]. β-sitosterol represents the main sterol in grapeseed oil, and α-tocopherol represents the main tocopherol, reaching values of around 70% of tocopherols, corresponding to 3.8 mg/100 g of oil [18]. Taking into account the incorporation of GP into the studied foods could be interesting from a nutritional point of view due to the incorporation of unsaturated fatty acids and tocopherols/tocotrienols which are important antioxidant lipid compounds and have been associated with health-promoting effects [40].

In the three GP-supplemented foods, decreases of 6, 13 and 14% of carbohydrate content were observed for muffins, cereal bars and biscuits, respectively. However, conventional and GP-supplemented foods are important sources of carbohydrates. In case of a potential intention of reducing even their content more, larger amounts of other ingredients could be replaced by GP.

As far as ash content is concerned, there was an increase in its values in the foods formulated with GP (90% in muffins, 105% in cereal bars and 79% in biscuits). This is in line with what was observed by Canett Romero et al. [41], who reported that the ash content in cookies supplemented with grape skins was higher due to the high mineral content of this by-product.

For the new products, an initial test of consumer’s acceptability of foods by a sensory analysis is necessary. As can be observed in Figure 1, the analysis showed that control foods had highest marks for all the sensory parameters, but, in general terms, GP-supplemented foods presented elevated marks also (between 3.71 and to 4.42), with GP-supplemented biscuits being the food which got the highest evaluation marks for all the sensory parameters (except for the “texture”). Figure 2 show photos of the baked foods with and without GP.

With respect to muffins, GP addition resulted in significant differences in all of the sensory attributes, except “taste”, compared to control. The highest difference was observed in the attribute “texture” (4.47 for control compared to 3.87 for supplemented muffins).

Focusing on cereal bars, it was observed that in attributes like “appearance”, “texture” and “color”, there were not significant differences following Fisher’s LSD test, but for “general opinion” and “taste” parameters, differences were observed, with “taste” showing the greatest difference.

Considering the biscuits, the attribute “texture” was the only parameter without significant differences between control and GP-supplemented foods, with “general opinion” being the attribute which showed significant differences in all the GP-supplemented foods compared to their control counterparts. To sum up, GP addition resulted in significant differences for most attributes for all tested foods, as compared to control. Although control foods had better marks compared to supplemented ones, the addition of GP powder to food products presented a notable acceptance by consumers.

## 3. Materials and Methods

### 3.1. Chemicals and Ingredients

Hydrochloric acid, ethanol and Folin–Ciocalteu reagent were purchased from Merck (São Paulo, Brazil). Trolox reagent (6-hydroxy-2,5,7,8-tetramethylchroman-2-carboxylic acid), NaH_2_PO_4_x2H_2_O, 2Na_2_HPO_4_x12H_2_O, fluorescein and 2,2′-azobis-2-methylpropionamidine dihydrochloride (AAPH) were purchased from Sigma-Aldrich (Steinheim, Germany). Ultrapure water was obtained from a Milli-Q system (Millipore, Billerica, MA, USA). Standards of gallic acid (99%), 3-hydroxytyrosol (≥99.5%), (−)-gallocatechin gallate (≥99%), caftaric acid (≥97%), chlorogenic acid (≥95%), (+)-catechin (≥99%), caffeic acid (99%), syringic acid (≥95%), coumaric acid (99%), ferulic acid (≥99%), trans-resveratrol (≥99%), quercetin hydrate (95%), quercetin 3-β-D-glucoside (≥90%), myricetin (≥96%), rutin trihydrate (99%) and malvidin-3-*O*-glucoside chloride (≥95%) were purchased from Sigma-Aldrich. Stock solutions of the abovementioned compounds were prepared in methanol at concentration levels of 1000 mg L^−1^. Calibration standards were dissolved in the initial mobile phase of each method (anthocyanins or non-anthocyanins). HPLC-grade acetonitrile (MeCN) and formic acid (FA) were acquired from Mallinckrodt Baker Inc. (Phillipsburg, NJ, USA). Primary–secondary amine (PSA) and octadecylsilane (C18) were both obtained from Waters (Milford, MA, USA). Reagent-grade NaCl, anhydrous Na_2_CO_3_, anhydrous MgSO_4_ and anhydrous CaCl_2_ were purchased from Sigma-Aldrich.

The bread wheat flour (0000 quality) used for baking manufacturing was bought from Molinos Rio de la Plata (Buenos Aires, Argentina). White sugar (common type A, purified and crystallized sucrose) was from Ingenio Bella Vista, Jose Minetti y Cia, Ltda. S.A.C.I. (Tucumán, Argentina). Butter (extra quality with salt) was from Mastellone Hnos S.A. (Buenos Aires, Argentina), d. Flavoring (synthetic vanilla solution) was purchased from Dulfix S.A. (Buenos Aires, Argentina). Baking powder was sourced from Mondelez (Buenos Aires, Argentina). Honey was provided by Ignacio Toledo Industry (Mendoza, Argentina). Oats (traditional), corn and rice flakes were provided by Granix S.A. (Buenos Aires, Argentina) and sunflower oil was from Molinos Rio de la Plata S.A. (Buenos Aires, Argentina).

### 3.2. Grape Pomace Powder Preparation

In this study, GP from red *Vitis vinifera* L. cv. Malbec was used. This was acquired from a local winery from several vineyards located in Gualtallary, Mendoza, Argentina. Fresh GP was collected immediately after pressing the grapes in the winery and placed in ice-cooled boxes during transportation to the laboratory and then stored at −20 °C until processing. The GP was dried at 60 °C for 48 h, then it was ground using a food processor (kitchen blender) and sifted using a 1 mm mesh to obtain a fine powder and stored at −20 °C until further use.

### 3.3. Foods Preparation

#### 3.3.1. Muffins

The formulation of the control muffins is listed in Table 5. A proportion of 5% of GP was used to replace a part of the wheat flour (*w*/*w*) to prepare the GP-supplemented muffins. 

The oven was conditioned with a pan of water and heated to 180 °C. All the dry ingredients were combined in a bowl. Afterwards, wet ingredients were also mixed. Then, both preparations were combined until the ingredients were completely integrated. The batter was portioned into 40 × 47 mm paper muffin liners and baked at 180 °C for 20 min. The muffins were cooled for 60 min and stored at −20 °C for further analysis.

#### 3.3.2. Biscuits

The biscuit formulation is presented in Table 5. In this case, 10% of GP instead of wheat flour (*w*/*w*) was used for GP-supplemented biscuits.

The oven was preheated to 180 °C and conditioned with a pan of water. The liquid ingredients were combined in a separate bowl, added to the dry ingredients and mixed. The batter was poured into a 30 cm × 10 cm rectangular baking pan and baked at 180 °C for 30 min. The sponge cake obtained was cut into slices which were baked again at 200 °C for 10 min for the roasting process. The biscuits were cooled in the pan for 60 min and stored at −20 °C until further use.

#### 3.3.3. Cereal Bars

An amount of 10% GP (*w*/*w*) was added to the GP-supplemented cereal bars formula (Table 5). To prepare the cereal bars, all ingredients were mixed and put into a rectangular pan with nonstick oven paper. The batter was baked at 160 °C for 15 min in a preheated oven. Then, the product obtained was cooled for 60 min and cut into 20 g bars (approximately 3 × 10 cm). Cereal bars were stored at −20 °C for further analysis. It is convenient to mention that a portion of the prepared foods was kept at room temperature until the sensory analysis.

It should be noted that in GP-supplemented muffins and biscuits, there was a substitution of a part of the wheat flour (*w*/*w*) with GP, while in the case of GP-supplemented cereal bars, GP powder was an extra ingredient. Common white wheat flour, sunflower oil, butter, eggs, vanilla, honey, oats, cornflakes, rice flakes, sugar and baking powder were purchased at local markets.

### 3.4. Phenolic Compounds Extraction

Solid–liquid extraction was applied to recover the PCs from food samples. Briefly, dried samples were ground in a laboratory mixer. Then, 5 mL of extraction solvent (ethanol–water in a 4:1 *v*/*v* ratio with 1% HCl) was added to tubes containing 1.5 g of sample powder. The mixture was sonicated for 30 min at 25 °C with manual shaking at 5 min intervals and then centrifuged at (4000 rpm) for 30 min. The supernatant was removed to a clean tube. The pellet was subjected to a second extraction step. The supernatants were put together, dried and stored at −20 °C until analysis.

### 3.5. Total Phenolic Content (TPC)

The TPC was spectrophotometrically measured with a Cary-50 UV–vis spectrophotometer (Varian Inc., Mulgrave, Australia). TPC was determined by the Folin–Ciocalteu assay (FC) according to Antoniolli et al. [5]. The extracts reacted with the Folin-Ciocalteu reagent for 5 min, then were added to 3.75 mL of 20% Na_2_CO_3_. The samples were incubated for 90 min in the dark at room temperature. The absorbance was measured at 750 nm using a Cary-50 spectrophotometer (Varian Inc., Mulgrave, Australia). A gallic acid standard calibration curve was used to determine the TPC. Results were reported as milligrams of gallic acid equivalents (mg GAE g^−1^) and then referenced based on dry matter. 

### 3.6. Antiradical Capacity (AC)

The ORAC was determined according to Antoniolli et al. [5]. Food extracts were diluted 1:750 *v*/*v* in 75 mmol L^−1^ potassium phosphate buffer (pH 7.0). Fifty microliters of diluted samples and Trolox standards (0, 3.125, 6.25, 12.5, 25 and 50 µmol L^−1^) were added to a 96-well plate. Then, 100 µL of fluorescein solution was added and the mixture was incubated at 37 °C for 7 min before the addition of 50 µL of 140 mmol L^−1^ of the peroxyl radical generator AAPH. Fluorescence intensity was monitored by using 485 nm excitation and 538 nm emission at 1 min intervals for 90 min on a microplate fluorometer (Fluoroskan Ascent FL, Thermo Fisher Scientific Inc., Wilmington, DE, USA). The area under the curve of the fluorescence decay during 90 min was calculated and the ORAC was expressed as µmol of Trolox equivalents per gram of food (µmol TE g^−1^ DW).

### 3.7. Non-Anthocyanins Phenolic Compounds

The identification and quantification of PCs present in the food samples were carried out by high-performance liquid chromatography coupled to diode array and fluorescence detectors (HPLC–DAD–FLD) using a Dionex Ultimate 3000 (Dionex Softron GmbH, Thermo Fisher Scientific Inc., Germering, Germany) equipped with a vacuum degasser unit, an autosampler, a quaternary pump, a chromatographic oven, a diode array (Dionex DAD-3000 (RS)) and a dual-channel fluorescence detector (FLD-3400RS Dual-PMT) connected in series. Chromeleon 7.1 software was used to control all the acquisition parameters of the HPLC–DAD–FLD system and to process the obtained data. 

Separation and quantification were carried out on a reversed-phase Kinetex C_18_ column (3.0 mm × 100 mm, 2.6 μm; Phenomenex, Torrance, CA, USA). The chromatographic conditions were similar to those previously used in our previous work [42]. An aqueous solution of 0.1% FA (A) and MeCN (B) were used as mobile phases. Analytes were separated using the following gradient: 0–1.7 min, 5% B; 1.7–10 min, 30% B; 10–13.5 min, 95% B; 13.5–15 min, 95% B; 15–16 min, 5% B; 16–19, 5% B. The total flow rate was set at 0.8 mL min^−1^. The column temperature was 35 °C and the injection volume was 5 μL. The analytical flow cell for DAD was set to scan from 200 nm to 400 nm. Wavelengths of 254 nm, 280 nm, 320 nm and 370 nm were used depending on the targeted analytes for DAD, while an excitation wavelength (Ex) of 290 nm and monitored emission (Em) responses of 315 nm, 360 and 400 nm were used depending on the analytes. The identification of PCs in samples was based on the comparison of the retention times (tR) in samples with those of authentic standards. Sample quantification was performed by external calibration with pure standards according to Ferreyra et al. [42]. Calibration plots for studied analytes showed linear ranges between 0.05 and 40 mg L^−1^ with coefficient of determination (r^2^) higher than 0.991.

Extracts of food samples were cleaned up by QuEChERS (quick, easy, cheap, effective, rugged and safe) coupled with dispersive solid-phase extraction (d-SPE) clean-up according to a previous report with some modifications [43]. The analytes were extracted from an aliquot of extracts of the food samples (previously acidified with 1% FA) using 2.5 mL MeCN. The rest of the protocol was as described by Fontana and Bottini [43]. An aliquot of the obtained cleaned extract was concentrated to dryness and taken up with the initial mobile phase prior to HPLC–DAD–FLD analysis.

### 3.8. Anthocyanins

The analysis of anthocyanins was performed with the same equipment and chromatographic column as for non-anthocyanins, following the method reported by Fontana et al. [44]. The mobile phases consisted of ultrapure water, FA and MeCN with 87:10:3 concentration, *v*/*v*/*v* (A) and ultrapure water, FA and MeCN with 40:10:50 concentration, *v*/*v*/*v* (B) using the following gradient: 0 min, 10% B; 0–6 min, 25% B; 6–10 min, 31% B; 10–11 min, 40% B; 11–14 min, 50% B; 14–15 min, 100% B; 15–17 min, 10% B; 17–21 min, 10% B. The mobile phase flow was 0.8 mL min^−1^, column temperature was 35 °C, and the injection volume was 10 μL. Quantification was carried out by measuring the area at 520 nm, and the anthocyanin content was expressed as malvidin-3-glucoside, using an external standard calibration curve (1–250 μg mL^−1^, r^2^ = 0.9984). Prior to sample analysis, extracts of foods obtained as described in Section 3.4 were dried by means of a SpeedVac concentrator and then resuspended in the initial mobile phase of the chromatographic method for anthocyanins.

### 3.9. Nutritional Parameters

In order to define the nutritional contribution of the product, a proximal chemical analysis was carried out by evaluating moisture, proteins, carbohydrates, lipids, fiber and ashes using official methods.

Moisture content was determined using AOAC official method 925.10 [45], by drying at 105 °C until the resulting weight was constant. The ash value was determined by the destruction of the organic matter present in the sample by calcination and gravimetric determination of the residue using a muffle at 550 °C until white ashes were observed, according to AOAC official method 923.03 [45].

The Kjeldahl method was used to determine protein content with N × 6.25 (N = total nitrogen) according the AOAC (method 979.06) [45], while the lipid content was tested using the Soxhlet extraction method, as described in AOAC official method 920.39 [45].

The determination of total dietary fiber was carried out by the enzymatic–gravimetric procedure as described in the AOAC (method 985.29) [45]. Analyses were conducted in triplicate on each formulation. The carbohydrate content was determined as the difference subtracting the protein, ash, moisture and lipid contents from 100.

### 3.10. Sensory Analysis

For the sensory evaluation, a test was carried out to obtain the degree of acceptance by potential consumers, using a hedonic scale of acceptance of the product. A total number of 45 untrained judges, habitual consumers, participated in the test. To gain a more comprehensive understanding of the product, personal data of consumers, including gender and age range, were collected. it was found that the majority of the population were women (71%). The remaining judges were male. The surveyed individuals ranged from 20 to 60 years old. Among them, 42% were between 31 and 40 years old, 33% were between 20 and 30 years old, 18% were in the range of 41 to 50 years old and finally, the smallest proportion of the judges, representing 7% of the sample, were aged between 51–60 years old.

The evaluated samples were the GP-supplemented foods. Control samples without the addition of GP were also given to the judges. The sensory analysis was conducted using a scale ranging from 1 to 5 to evaluate various attributes, including “visual appearance”, “color”, “taste”, “texture” and “general opinion”. A rating of 5 represented “like extremely”, while a rating of 1 indicated “dislike extremely” [23]. In addition, consumers answered whether or not they were regular consumers of these types of foods and whether or not they would buy the products and the reason for that potential purchase decision. The samples were served in plastic cups. Consumers were informed about the addition of GP to the food product. Sensory statistical analysis was performed by analysis of variance (ANOVA) using Infostat v 2022 [46]. Variables were considered significant with *p* < 0.05 and differences were determined based on Fisher’s least significant difference test (LSD).

## 4. Conclusions

The present study presented evidence of the potentiality of the fortification of different baked products with cv. Malbec GP to improve the antioxidant fiber and PCs content, aiming to improve the foods’ functionality. The incorporation of GP increased the content of dietary fiber, proteins, ash, TPC, AC, anthocyanins and non-anthocyanin PCs. In terms of incorporation of the PCs into the foods supplemented with GP, one portion of the supplemented foods showed levels high enough to satisfy the recommended dose per day of these bioactive compounds. Also interesting was the high acceptability of the foods during the sensory evaluation, which needs to be complemented with more in-depth descriptors in the future. The detailed PCs profiling in these foods evidenced a group-specific anthocyanins instability, observing that the acylated derivatives were more stable under heat treatment (baking) in food processing compared to non-acylated counterparts. Therefore, the direct utilization of GP gives to these bakery foods nutritionally enhanced properties, encouraging their use as sustainable sources of functional components for the food industry in a growing context of circular economy.

## Figures and Tables

**Figure 1 plants-13-00590-f001:**
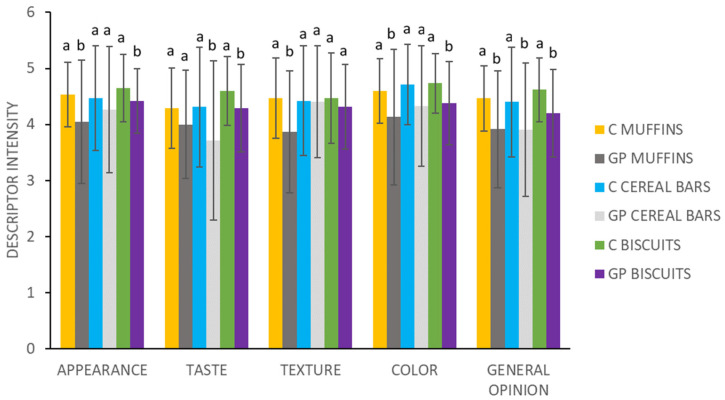
Overall sensory quality of control (designed as C) and GP-supplemented foods. Scale from 1 (extreme dislike) to 5 (extreme like). For each attribute and type of food, bars with different letters are significantly different (*p* < 0.05) following Fisher’s LSD test.

**Figure 2 plants-13-00590-f002:**
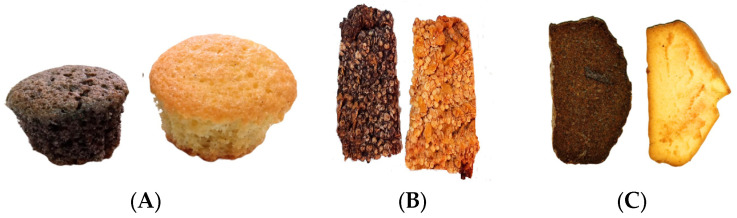
(**A**) Muffins (**B**), cereal bars and (**C**) biscuits. For each photo of the foods, the left corresponds to the GP-supplemented food and the right shows the corresponding control.

**Table 1 plants-13-00590-t001:** TPC and AC of GP-enriched and control biscuits, muffins and cereal bars. Average values with their standard deviations, n = 3 replicates.

Sample	GP Muffin	Control Muffin	GP Cereal Bar	Control Cereal Bar	GP Biscuit	Control Biscuit
TPC (mg GAE g^−1^ DW)	3.28 ± 0.19 ^a^	0.39 ± 0.04 ^b^	6.27 ± 0.19 ^a^	2.58 ± 0.15 ^b^	6.35 ± 0.20 ^a^	0.94 ± 0.10 ^b^
AC (μmol TE g^−1^ DW)	47.84 ± 1.55 ^a^	7.22 ± 0.03 ^b^	75.40 ± 6.77 ^a^	24.00 ± 0.49 ^b^	68.32 ± 4.86 ^a^	15.43 ± 0.45 ^b^

Different letters indicate significant differences. For each compound and type of food, columns with different letters are significantly different (*p* < 0.05), comparing the GP-supplemented food and control following Fisher’s LSD test.

**Table 2 plants-13-00590-t002:** Individual anthocyanins quantification on GP-supplemented foods. Average contents (µg g^−1^ WM of GP-supplemented foods) with their standard deviations, n = 3 replicates.

Anthocyanin	Muffin	Cereal Bar	Biscuit
Delphinidin 3-*O*-glucoside	14.1 ± 0.3 ^b^	28.8 ± 0.9 ^a^	10.4 ± 2.9 ^b^
Cyanidin 3-*O*-glucoside	8.2 ± 0.1 ^a^	6.3 ± 0.1 ^a^	4.9 ± 0.1 ^b^
Petunidin 3-*O*-glucoside	20.2 ± 0.7 ^b^	36.9 ± 1.1 ^a^	9.8 ± 0.1 ^c^
Peonidin 3-*O*-glucoside	10.3 ± 0.1 ^a^	13.8 ± 0.3 ^a^	5.5 ± 0.1 ^b^
Malvidin 3-*O*-glucoside	160.2 ± 4.1 ^b^	224.9 ± 8.9 ^a^	69.7 ± 1.6 ^c^
Total glycosylated	212.9	310.7	100.5
Delfinidin 3-*O*-acetylglucoside	8.9 ± 0.1 ^a^	11 ± 0.2 ^a^	7.4 ± 0.2 ^c^
Peonidin 3-*O*-acetylglucoside	60.4 ± 1.9 ^b^	80.8 ± 3.1 ^a^	45.2 ± 0.0 ^c^
Total acetylated	69.3	91.8	52.6
Cyanidin 3-*O*-p-coumaroylglucoside	13.8 ± 3.2 ^a^	16.5 ± 3.7 ^a^	7.8 ± 0.5 ^b^
Peonidin 3-*O*-p-coumaroylglucoside	39.1 ± 1.3 ^b^	50.1 ± 1.7 ^a^	25.8 ± 0.2 ^c^
Malvidin 3-*O*-p-coumaroylglucoside	558.1 ± 31.1 ^b^	695.8 ± 34.7 ^a^	361.5 ± 4.3 ^c^
Total coumaroylated	611.1	762.3	395.1
Total anthocyanins	893.2	1164.8	548.2

Different letters indicate significant differences. For each compound and type of food, columns with different letters are significantly different (*p* < 0.05) following Fisher’s LSD test.

**Table 3 plants-13-00590-t003:** Non-anthocyanins composition of GP-supplemented foods. Average contents (µg g^−1^ WM of GP-supplemented food) with their standard deviations, n = 3 replicates.

	GP Muffin	Control Muffin	GP Cereal Bar	Control Cereal Bar	GP Biscuit	Control Biscuit
Hydroxybenzoic acids						
Syringic acid	55.4 ± 2.7 ^a^	n.d. ^b^	45.5 ± 1.8 ^a^	4.4 ± 1.3 ^b^	107.8 ± 2.4 ^a^	n.d. ^b^
Total	55.4		45.5	4.4	107.8	
Hydroxycinnamic acids						
Caffeic acid	60.6 ± 3.03 ^a^	n.d. ^b^	48.4 ± 1.8 ^a^	20.8 ± 2.0 ^b^	99.3 ± 3.5 ^a^	6.8 ± 0.8 ^b^
*p*-coumaric acid	35.2 ± 1.7 ^a^	n.d. ^b^	23.2 ± 1.1 ^a^	n.d. ^b^	35.0 ± 1.5 ^a^	n.d. ^b^
Ferulic acid	3.8 ± 0.2 ^a^	n.d. ^b^	6.5 ± 0.1 ^a^	n.d. ^b^	10.0 ± 0.4 ^a^	n.d. ^b^
Chlorogenic acid	29.6 ± 1.5 ^a^	n.d. ^b^	121.8 ± 0.1 ^a^	n.d. ^b^	n.d.	n.d.
Total	129.2		199.9	20.8	144.3	6.8
Stilbenes						
*trans*-resveratrol	2.7 ± 0.1 ^a^	n.d. ^b^	2.0 ± 0.1 ^a^	n.d. ^b^	4.5 ± 0.1 ^a^	n.d. ^b^
Total	2.7		2.0		4.5	
Flavanols						
(+)-catechin	44.8 ± 2.2 ^a^	n.d. ^b^	60.1 ± 1.3 ^a^	n.d. ^b^	51.6 ± 3.5 ^a^	n.d. ^b^
Total	44.8		60.1		51.6	
Flavonols						
Quercetin-3-*O*-glucoside	103.7 ± 5.2 ^b^	126.6 ± 6.3 ^a^	11.0 ± 1.7 ^b^	44.3 ± 8.2 ^a^	194.8 ± 1.7 ^b^	285.0 ± 5.3 ^a^
Quercetin	n.d.	n.d.	4.7 ± 0.8 ^a^	4.0 ± 0.2 ^a^	3.9 ± 0.2 ^a^	n.d. ^b^
Myricetin	4.2 ± 0.2 ^a^	n.d. ^b^	7.6 ± 0.2 ^b^	5.4 ± 0.2 ^a^	8.0 ± 1.1 ^a^	n.d. ^b^
Rutin	62.8 ± 3.1	4.1 ± 0.2	51.1 ± 1.2	3.4 ± 0.3	80.5 ± 0.5	n.d.
Total	170.7	130.7	74.4	57.1	287.2	285.0
Other compounds						
OH-tyrosol	8.6 ± 0.4 ^a^	n.d. ^b^	6.9 ± 0.6 ^a^	n.d. ^b^	13.3 ± 1.2 ^a^	n.d. ^b^
Total	8.6		6.9		13.3	
Total PCs	664.8	293.5	442.0	186.7	771.0	350.0

Different letters indicate significant differences. For each compound and type of food, columns with different letters are significantly different (*p* < 0.05) following Fisher’s LSD test. n.d.: not detected.

**Table 4 plants-13-00590-t004:** Proximate composition of GP-supplemented foods (g/100 g FW). Average contents with their standard deviations, n = 3 replicates.

Sample	GP Muffin	Control Muffin	GP Cereal Bar	Control Cereal Bar	GP Biscuit	Control Biscuit
Moisture	21.51 ± 0.32 ^b^	20.25 ± 0.17 ^a^	20.54 ± 0.51 ^a^	14.31 ± 0.81 ^b^	6.13 ± 0.28 ^a^	3.55 ± 0.01 ^b^
Proteins	7.33 ± 0.12 ^a^	6.6 ± 1.15 ^a^	4.67 ± 0.29 ^a^	4.13 ± 0.32 ^a^	11.27 ± 0.72 ^a^	11.3 ± 0.69 ^a^
Lipids	20.33 ± 0.47 ^a^	21.33 ± 0.64 ^a^	1.17 ± 0.25 ^a^	0.70 ± 0.14 ^b^	14.20 ± 0.30 ^a^	12.40 ± 0.62 ^b^
Fiber	1.9 ± 0.17 ^a^	0.32 ± 0.09 ^b^	3.31 ± 0.43 ^a^	0.49 ± 0.08 ^b^	5.83 ± 1.59 ^a^	0.90 ± 0.26 ^b^
Carbohydrates	47.7 ± 0.35 ^b^	50.89 ± 0.92 ^a^	69.48 ± 0.97 ^b^	80.18 ± 1.49 ^a^	60.65 ± 0.43 ^b^	70.64 ± 1.40 ^a^
Ash	1.22 ± 0.05 ^a^	0.68 ± 0.15 ^b^	0.84 ± 0.06 ^a^	0.40 ± 0.02 ^b^	1.91 ± 0.27 ^a^	1.21 ± 0.18 ^b^

Different letters indicate significant differences. For each parameter and type of food, columns with different letters are significantly different (*p* < 0.05), comparing the GP-supplemented food and control following Fisher’s LSD test.

**Table 5 plants-13-00590-t005:** Formulation for control muffin, biscuit and cereal bars.

Muffin (% *w*/*w*)	Cereal Bars (% *w*/*w*)	Biscuit (% *w*/*w*)
Flour: 24	Honey: 63	Flour: 25
Egg: 27	Oats: 17.5	Egg: 46
Sugar: 27	Cornflakes: 10.5	Sugar: 23
Butter: 21	Rice flakes: 9	Sunflower Oil: 5
Vanilla: 0.5		Vanilla: 0.5
Baking Powder: 0.5		Baking Powder: 0.5

## Data Availability

The data will be available by contacting the corresponding author.

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
