# Peer review of "Phenolic, Nutritional and Sensory Characteristics of Bakery Foods Formulated with Grape Pomace"

_plants, 2024, doi:10.3390/plants13050590_

Round 1

Reviewer 1 Report

Comments and Suggestions for Authors

Why did you only analyze muffins with 5% of GP

For biscuits, why did you use a percentage of 10%?

For the cereal bars you used a percentage of 10%?

For all three products, I recommend trying with higher percentages of GP, to observe what happens with the biologically active compounds, considering that the GP from red grapes is studied and has a large intake of phytochemical compounds.

Author Response

Why did you only analyze muffins with 5% of GP

For biscuits, why did you use a percentage of 10%?

For the cereal bars you used a percentage of 10%?

-The decision of the amount of GP supplementation for each food was based on two factors. 1-previous studies evaluating the clinical effects of GP addition. For example, Martínez-Maqueda et.al. 2018 and Ramos-Romero et al. 2020 supplied 8 g of dried GP per day. In that way, we tried to select the supplemented amount based on the estimated quantity of PCs and dietary fiber on the final product to have a positive effect on health and follow diet recommendations of these bioactive components. In this work, the objective was to prepare foods for which at less two portions achieve the recommended amount of bioactive compounds. The other consideration was based on preliminary assays that we made, where we evaluated different food formulations selecting those with better technological and sensorial properties. In the case of muffins, when an amount higher than 5 % of GP was added, the volume and texture of the final product were severely affected (See new Fig. 2). For the other foods, we were able to work with higher amounts without having several effects on these properties.

Martínez-Maqueda, D., Zapatera, B., Gallego-Narbón, A., Vaquero, M. P., Saura-Calixto, F., & Pérez-Jiménez, J. (2018). A 6-week supplementation with grape pomace to subjects at cardiometabolic risk ameliorates insulin sensitivity, without affecting other metabolic syndrome markers. Food & function, 9(11), 6010-6019.

Ramos-Romero, S., Martínez-Maqueda, D., Hereu, M., Amézqueta, S., Torres, J. L., & Pérez-Jiménez, J. (2020). Modifications of gut microbiota after grape pomace supplementation in subjects at cardiometabolic risk: A randomized cross-over controlled clinical trial. Foods, 9(9), 1279.

For all three products, I recommend trying with higher percentages of GP, to observe what happens with the biologically active compounds, considering that the GP from red grapes is studied and has a large intake of phytochemical compounds.

-We thank the reviewer comment and we will take into account this consideration for future studies. We agree and know about the importance of the interesting phytochemical composition of GP, but it is also known that too high concentrations of GP also can cause less acceptability of the product due to the high content of astringent compounds like procyanidins. In addition, negative changes of texture and bakery quality of the foods need to be taken in mind as we observed in preliminary studies (also can be observed in the Fig. 2).

Reviewer 2 Report

Comments and Suggestions for Authors

Article

Phenolic, nutritional and sensory characteristics of functional foods formulated with grape pomace

A brief summary

The topic is important, interesting and in keeping with current trends of zero waste in horticulture production. The research, in general, is properly designed, the results interesting. However, corrections and additions are required.

Broad comments

1. The main deficiency of the presented work is the lack of statistical analysis in the first experiments and the not entirely correct interpretation of the results of the statistical analysis in the last study. The relevant tables should be supplemented with the sample size (number of replicates), the name and parameters of the statistical analysis method used and the results of this analysis.

a. The results for TPC, AA, individual anthocyanins, non-anthocyanin compounds and nutritional parameters were not statistically analysed. Therefore, conclusions such as those in lines 95-96, 186-194, 351-253, 261-271 or 279-300 are made without providing evidence in the form of statistical test results and are therefore not valid.

c. The results of the sensory test were in fact statistically analysed but, despite the information in the description of Figure 1, were not presented with the results of these tests. The high value of the standard deviation, on the other hand, suggests that the differences were not statistically significant, a fact that was not entirely properly described and interpreted in section 2.4. The question is also what was tested statistically. Was a one-way ANOVA conducted? What was compared with the control (prefix C)?  Were tests performed for individual traits or for product types?

2. Why weren't all these analyses carried out directly on the grape pomace? Before being added to the baked goods? A comparison of the content of the tested compounds in the plant material and in the finished cakes would provide answers to many of the questions raised by the authors themselves (e.g. lines 142-164).

3. The drying description is incomplete. What method was used for drying? What type of dryer? Was the material defrosted first? How? Is grinding the dried plant material with a kitchen blender the correct method?

4. It seems that standardising the presentation of the data would improve the readability of the article. At the moment, in Tables 1 and 5 the cakes are in columns and in Tables 2, 3 and 4 in rows.

5. The 'Literature' chapter needs to be revised and adapted to citation standards, e.g. long or short journal names or years of publication.

Specific comments

Table 1 & 2. It is likely that the order of the table has been swapped – see line 94, 371, 379.

Figure 1. Both parts (A) and (B) present the same data. They only differ in the way they are presented. You should decide on one type of data presentation and not duplicate it.

Author Response

Article

Phenolic, nutritional and sensory characteristics of functional foods formulated with grape pomace

A brief summary

The topic is important, interesting and in keeping with current trends of zero waste in horticulture production. The research, in general, is properly designed, the results interesting. However, corrections and additions are required.

Broad comments

  1. The main deficiency of the presented work is the lack of statistical analysis in the first experiments and the not entirely correct interpretation of the results of the statistical analysis in the last study. The relevant tables should be supplemented with the sample size (number of replicates), the name and parameters of the statistical analysis method used and the results of this analysis.

-We thank the reviewer comment and agree with the importance of statistical analysis for the correct interpretation of the results. We modified the tables adding the required information in those where it was not present in the submitted original version.

  1. The results for TPC, AA, individual anthocyanins, non-anthocyanin compounds and nutritional parameters were not statistically analysed. Therefore, conclusions such as those in lines 95-96, 186-194, 351-253, 261-271 or 279-300 are made without providing evidence in the form of statistical test results and are therefore not valid.

-As was commented before, we added the statistical analysis where they were missing and revised the discussions/conclusions achieved in those topics to validate the results. We also checked for some extra mistakes calculations that were properly revised.

  1. The results of the sensory test were in fact statistically analysed but, despite the information in the description of Figure 1, were not presented with the results of these tests. The high value of the standard deviation, on the other hand, suggests that the differences were not statistically significant, a fact that was not entirely properly described and interpreted in section 2.4. The question is also what was tested statistically. Was a one-way ANOVA conducted? What was compared with the control (prefix C)?  Were tests performed for individual traits or for product types?

-We really apologize for the Figure 1. We make a mistake during copy and paste the Fig. 1 from the original software to the template of the journal and missed the letters showing the significance of the analysis. Sorry again for that silly mistake and for not properly checking the Figure before submitting the manuscript. Now, the figure is right and have the complete information as we were intended to present in the first version of the paper. On the other side, we understand that standard deviations are high, but that is more or less normal in a subjective test like the performed. The information obtained is preliminary and other sensory analysis need to be performed in the future to validate the acceptability with a descriptive sensory analysis, for example. Anyway, the significance that we wrote in the text is that we observed with the performed analysis.

To clarify the comparison, as was written in the Fig. 1 caption, for each attribute and type of food, bars with different letters are significantly different (p < 0.05) from the control (C). Just to be more specific, i.e. for “appearance” attribute in muffins, the comparison was performed between C and GP supplemented muffins. Another individual comparison for control and GP supplemented was performed for cereal bars and biscuits, respectively. So, the comparison was only for the pars C and supplemented of each food as was originally discussed in the paper.

  1. Why weren't all these analyses carried out directly on the grape pomace? Before being added to the baked goods? A comparison of the content of the tested compounds in the plant material and in the finished cakes would provide answers to many of the questions raised by the authors themselves (e.g. lines 142-164).

-We agree with the comment of the reviewer and understand the importance of directly analyze the grape pomace when the research is focused on it profile. We already did these investigations in previous works and the focus here was to analyze the composition and stability of bioactive compounds in the supplemented foods. In fact, we discussed the anthocyanins changes compared to our previous publication in terms of the distribution of anthocyanin derivatives and properly justified our hypothesis based on the different stability of derivatives. Unfortunately, after checking our back-up samples we realized that any aliquot of the original grape pomace used for foods supplementation was remaining to satisfy the requirement of the reviewer to re-analyze a GP sample. In this context, we cannot analyze a sample of grape pomace.

  1. The drying description is incomplete. What method was used for drying? What type of dryer? Was the material defrosted first? How? Is grinding the dried plant material with a kitchen blender the correct method?

-Frozen grape pomace was defrosted and then dried for 48 h at 60 °C in a laboratory oven as described by Nakov et al. 2020. The use of a kitchen blender gives us the possibility to process enough amount of material with an adequate granulometry for food supplementation with GP and was also used in previous research. The option of using laboratory scale blenders was avoided because the equipment’s that we have in our laboratory are too small to process the amount of GP that we were needing for the experiments of food supplementation. In addition, as was written in the original text, after the grinding, the GP was sifted using a 1 mm mesh to obtain a fine powder. In this sense, for the objective of the study, we think that the use of a kitchen blender is an adequate, cheap and representative approach.

  1. It seems that standardising the presentation of the data would improve the readability of the article. At the moment, in Tables 1 and 5 the cakes are in columns and in Tables 2, 3 and 4 in rows.

-We agree with the reviewer and the presentation of the data was improved taking into account the comment. Tables 1 and 5 are now presented in a similar format of Tables 2, 3 and 4.

  1. The 'Literature' chapter needs to be revised and adapted to citation standards, e.g. long or short journal names or years of publication.

-We revised the literature to have a homogeneous format and agreeing with the journal requirements.

Specific comments

Table 1 & 2. It is likely that the order of the table has been swapped – see line 94, 371, 379.

-We apologize for the mistake. It was properly revised overall the manuscript.

Figure 1. Both parts (A) and (B) present the same data. They only differ in the way they are presented. You should decide on one type of data presentation and not duplicate it.

-We agree that the same data is presented. We decided to keep the part B and change the colors according to the suggestion of reviewer 3. Thanks for the suggestion to improve the presentation quality of the paper.

Reviewer 3 Report

Comments and Suggestions for Authors

plants-2853883

Title: “Phenolic, nutritional and sensory characteristics of functional foods formulated with grape pomace”

The aim of the study was to investigate the effect of the addition of grape pomace, obtained as a by-product of wine making, on the health and sensory properties of bakery products enriched with it. The objective of the work was achieved mainly through simple tests that have been used for years, although the methodology is well thought out and includes tests necessary at this stage of analysis.

The Authors used the term ‘functional food’ in the title. I would be careful about naming the products received this way. Although the additives used are indeed very valuable ingredients containing polyphenols and fiber, the "remaining" part of the product may in this case have a harmful effect on health, which means that the entire final product will be difficult to classify as health-beneficial. At a time when many millions of people suffer from type II diabetes and related serious vascular complications, promoting confectionery products containing sugar (even at 27% in the case of muffin mass) and refined white flour may be harmful to most of us. To sum up this argument, I naturally appreciate the Authors' efforts to improve existing products, but using the term ‘functional food’ in their case is too careless and unbalanced. Therefore, in my opinion, the title should be modified, while the keyword "bakery functional food" can be left, because the work fits into the trends of creating this type of food. This is a minor change in the title, but of great importance, which should result in a modification of the Authors' approach to the tested products in the Introduction, Discussion and Conclusions.

Apart from the above, I do not have many comments on the work, most of them are minor shortcomings and suggestions for improving the manuscript itself. I have listed them below.

1. Currently, when the content of antioxidants is tested, even if their changes over time are recorded, as in the ORAC method, the term antioxidant capacity or, more precisely, antiradical capacity is used. This is the case in the vast majority of recent works. Activity and its units are considered kinetic quantities. The expansion of the abbreviation ORAC itself has the word ‘capacity’. I would consider changing the term ‘antioxidant activity’.

2. The Authors titled section 2.1 as "TPC and AA" and 3.2  as “GP powder preparation”. The section titles should be independent of the text and contain complete, fully explanatory names.

3. In the Table 2, the description of flour is imprecise. Wheat flour has several types, differing in dietary fiber and protein content and, therefore, in their intended use. There are seven of them in my country and each one is described with a different numerical value. Flours purchased in stores are also marked according to this system.

4. What type of sugar did the authors use? Was it saccharose? The type of sugar needs to be added. Also, a type of honey. Was it large-flowered honey, the most commonly used? But maybe linden, honeydew, heather, acacia or something else? Honey differs greatly in the content of polyphenols, so such supplementation is advisable. Moreover, the Authors using vanilla did not write whether it was a dried pod or synthetic vanilla (vanillin). In short, the description of the ingredients of individual bakery products needs to be supplemented.

5. I suggest analyzing the HPLC data for (–)-gallocatechin. Such large and varied standard deviations may indicate incorrect peak identification.

6. The Authors wrote: The formulation of the control muffin is listed in Table 1 (line 371). That is a mistake. The formulations are presented in Table 2. The Authors should check whether the remaining Tables are presented correctly in the text.

7. There are two ways of presenting percentages at work. In many places the percent symbol is preceded by a space, in others it is not (for example, compare lines: 107, 112, 137, 147 (twice)). The entire text should be standardized.

8.    In the presentation of chemical substances, the names of salts should be supplemented with multiplication signs before the number of water molecules (degree of hydration), a comma should be placed between the names of two salts, and the full name of AAPH begins with 2,2'-, not 2,20-.

9. In line 362, the Latin name (Vitis vinifera) should be italized. Cultivar names are usually written in single quotation marks.

10. If possible, I would use more varied (in terms of brightness) colors in Fig. 1A. On a black and white print, the lines are indistinguishable and the legend is barely visible.

11. Maybe the Authors have photos of tested products supplemented with GP and the control products? Such photos would make the article more attractive.

Feb. 5th, 2024

Author Response

plants-2853883

Title: “Phenolic, nutritional and sensory characteristics of functional foods formulated with grape pomace”

The aim of the study was to investigate the effect of the addition of grape pomace, obtained as a by-product of wine making, on the health and sensory properties of bakery products enriched with it. The objective of the work was achieved mainly through simple tests that have been used for years, although the methodology is well thought out and includes tests necessary at this stage of analysis.

The Authors used the term ‘functional food’ in the title. I would be careful about naming the products received this way. Although the additives used are indeed very valuable ingredients containing polyphenols and fiber, the "remaining" part of the product may in this case have a harmful effect on health, which means that the entire final product will be difficult to classify as health-beneficial. At a time when many millions of people suffer from type II diabetes and related serious vascular complications, promoting confectionery products containing sugar (even at 27% in the case of muffin mass) and refined white flour may be harmful to most of us. To sum up this argument, I naturally appreciate the Authors' efforts to improve existing products, but using the term ‘functional food’ in their case is too careless and unbalanced. Therefore, in my opinion, the title should be modified, while the keyword "bakery functional food" can be left, because the work fits into the trends of creating this type of food. This is a minor change in the title, but of great importance, which should result in a modification of the Authors' approach to the tested products in the Introduction, Discussion and Conclusions.

-We thank the kind recommendation of the reviewer and we understand the point. In this sense, we modified the title to avoid using the term “functional food”. The title is now: “Phenolic, nutritional and sensory characteristics of bakery foods formulated with grape pomace”. Resulting of that comment we revised the rest of the manuscript to be consistent.

Apart from the above, I do not have many comments on the work, most of them are minor shortcomings and suggestions for improving the manuscript itself. I have listed them below.

  1. Currently, when the content of antioxidants is tested, even if their changes over time are recorded, as in the ORAC method, the term antioxidant capacity or, more precisely, antiradical capacity is used. This is the case in the vast majority of recent works. Activity and its units are considered kinetic quantities. The expansion of the abbreviation ORAC itself has the word ‘capacity’. I would consider changing the term ‘antioxidant activity’.

-We thank the comment and the clear explanation of the reviewer. We changed overall the manuscript antioxidant activity by antiradical capacity.

  1. The Authors titled section 2.1 as "TPC and AA" and 3.2  as “GP powder preparation”. The section titles should be independent of the text and contain complete, fully explanatory names.

-We changed the title according to the suggestion.

  1. In the Table 2, the description of flour is imprecise. Wheat flour has several types, differing in dietary fiber and protein content and, therefore, in their intended use. There are seven of them in my country and each one is described with a different numerical value. Flours purchased in stores are also marked according to this system.

-We thank the reviewer comment. We added the description of ingredients used for foods elaboration in the section 3.1. The type of wheat flour utilized is described now in that sectionas it is regulated and classified in the Argentinian Food Code (Chapter IX).

  1. What type of sugar did the authors use? Was it saccharose? The type of sugar needs to be added. Also, a type of honey. Was it large-flowered honey, the most commonly used? But maybe linden, honeydew, heather, acacia or something else? Honey differs greatly in the content of polyphenols, so such supplementation is advisable. Moreover, the Authors using vanilla did not write whether it was a dried pod or synthetic vanilla (vanillin). In short, the description of the ingredients of individual bakery products needs to be supplemented.

-We thank the reviewer comment and completely agree with the importance to describe the ingredients used. As we commented bellow, we added the details of the ingredients used in section 3.1.

Specifically, the type of sugar (sucrose) utilized is described in section 3.1 as is regulated and classified in Argentinian Food Code (Chapter X). We make a translation mistake in the original version by using “saccharose”, the correct name is sucrose. Related to the honey, the identification of the floral type of honey is not mandatory in Argentina according to the Argentine Food Code. The vanilla ingredient correspond to synthetic vanilla (called vanillin) solution. In any case, we added the description of the ingredients of individual bakery products as the reviewer required.

  1. I suggest analyzing the HPLC data for (–)-gallocatechin. Such large and varied standard deviations may indicate incorrect peak identification.

-We really thank the reviewer comment. After analyzing the chromatographic data, for control and GP supplemented samples as well as checking the maximum absorption of the peak in each condition, we realized that it was identified erroneously. In that way, we eliminated the compound from the paper. We apologize for the mistake.

  1. The Authors wrote: The formulation of the control muffin is listed in Table 1 (line 371). That is a mistake. The formulations are presented in Table 2. The Authors should check whether the remaining Tables are presented correctly in the text.

-Thanks for the comment. We checked the other tables and now are correct. Sorry for the mistake.

  1. There are two ways of presenting percentages at work. In many places the percent symbol is preceded by a space, in others it is not (for example, compare lines: 107, 112, 137, 147 (twice)). The entire text should be standardized.

-Thanks for the comment. We revised and corrected to be uniform.

  1. In the presentation of chemical substances, the names of salts should be supplemented with multiplication signs before the number of water molecules (degree of hydration), a comma should be placed between the names of two salts, and the full name of AAPH begins with 2,2'-, not 2,20-.

-The recommendations of the reviewer were taken and changes made.

  1. In line 362, the Latin name (Vitis vinifera) should be italized. Cultivar names are usually written in single quotation marks.

-We put in italics. It was a formatting mistake.

  1. If possible, I would use more varied (in terms of brightness) colors in Fig. 1A. On a black and white print, the lines are indistinguishable and the legend is barely visible.

-We changed the colors of the figure to make more attractive and easily to distinguish the variables.

  1. Maybe the Authors have photos of tested products supplemented with GP and the control products? Such photos would make the article more attractive.

-We added the Fig. 2 showing the control and GP supplemented foods. The suggestion was really good also to explain some features related to the proportion of grape we decided to add.

Round 2

Reviewer 2 Report

Comments and Suggestions for Authors

If articles submitted for publication were always this well-written, reviewers would have very little to do.  This article is ready for publication.

Author Response

Thanks for the comment.

Reviewer 3 Report

Comments and Suggestions for Authors

The manuscript "Phenolic, nutritional and sensory characteristics of bakery foods formulated with grape pomace" has been revised and all my comments have been taken into account. Basically, I have no significant comments on the paper.

Reading the revised version, another thought occurred to me. Where the Authors describe the group of people who evaluated the bakery products, it would be good if they wrote whether the people were from different age groups. This is important because older people are more used to traditional flavors. If the Authors had conducted the test among students, as is common among university researchers, they probably would have gotten significantly different results. Therefore, it would be good to add a sentence describing the group of testers in some way.

I found a few more minor editing errors:

1. In Fig. 2, the letters describing bakery products overlap the text from line 347.

2. In line 369, the % symbol is written differently than in the entire work, i.e. without space.

3.  In line 395, the temperature value and unit should be placed in a single line.

4. In Table 4, I would standardize the line thickness.

Feb. 16, 2024

Author Response

The manuscript "Phenolic, nutritional and sensory characteristics of bakery foods formulated with grape pomace" has been revised and all my comments have been taken into account. Basically, I have no significant comments on the paper.

Reading the revised version, another thought occurred to me. Where the Authors describe the group of people who evaluated the bakery products, it would be good if they wrote whether the people were from different age groups. This is important because older people are more used to traditional flavors. If the Authors had conducted the test among students, as is common among university researchers, they probably would have gotten significantly different results. Therefore, it would be good to add a sentence describing the group of testers in some way.

-We thank the reviewer comment and agree with the importance of these information. We added these data on the revised manuscript.

I found a few more minor editing errors:

  1. In Fig. 2, the letters describing bakery products overlap the text from line 347.

-It was revised.

  1. In line 369, the % symbol is written differently than in the entire work, i.e. without space.

-It was corrected.

  1. In line 395, the temperature value and unit should be placed in a single line.

-This is not possible to do because of the predetermined template of the journal. This details probably should be corrected by the editorial office after acceptation of the paper.

  1. In Table 4, I would standardize the line thickness.

-We checked the Tables, and all of them have standardized line thickness when you check the pdf document. Is something related to the word template of the journal that make that lines are not looking the same thickness when you see the word document.